# The Price of the Ticket: Health Costs of Upward Mobility among African Americans

**DOI:** 10.3390/ijerph17041179

**Published:** 2020-02-13

**Authors:** Darrell Hudson, Tina Sacks, Katie Irani, Antonia Asher

**Affiliations:** 1Brown School of Social Work, Washington University, St. Louis, MO 63130, USA; katherineirani@wustl.edu; 2School of Social Welfare, University of California at Berkeley, Berkeley, CA 94720, USA; tsacks@berkeley.edu; 3School of Public Health, Tulane University, New Orleans, LA 70112, USA; aasher4@tulane.edu

**Keywords:** African American, socioeconomic status, social mobility, racism

## Abstract

There is a growing literature that has documented diminishing health returns on upward social mobility among Black Americans. Due to historical policies and practices, upward social mobility is often an arduous, isolating process for Black Americans, especially as they navigate predominately white educational and workplace settings. This paper advances the literature in several meaningful and innovative ways. The goal of this paper is to provide a qualitative account of the health costs of upward social mobility and describe how these costs could diminish health returns despite greater levels of socioeconomic resources. Focus groups and surveys were the data collection methods for the study. Inclusion criteria for the study were that respondents identified as African American or Black, were 24 years or older and had completed college. The total sample was 32 college-educated Black men (*n* = 12) and women (*n* = 20). The mean age for men was 39 (range = 26–50) and 33 years of age (range = 24–59) for women. Key findings highlighted in this paper include (1) hypervisibility and subsequent vigilance; (2) uplift stress; and (3) health costs associated with social mobility. The sum of these stressors is posited to affect multiple health outcomes and elucidate the mechanisms through which socioeconomic returns on health are diminished.

## 1. Introduction

Du Bois (1903) characterized the experience of Black people in America as one of “double consciousness” in which Black people come to view themselves from the perspective of white society [1]. Du Bois (1903) notes, “It is a peculiar sensation, this double-consciousness, this sense of always looking at one’s self through the eyes of others, of measuring one’s soul by the tape of a world that looks on in amused contempt and pity” (p. 3). This perpetual measurement is taxing, physically and psychologically, and may be even more acute for middle-class Black people, who navigate invisible boundaries and cross barriers, especially in public spaces such as the workplace. Du Bois’ analysis of the interlocking systems of class and racial exploitation inform analyses of the vulnerability of upwardly mobile Black Americans.

Although DuBois is perhaps most well-known for the notion of “double consciousness”, his work also highlighted prejudice as a constituent element in the reproduction of systematic racial inequality (Bobo, 2000). Moreover, he outlined the effects of interpersonal prejudice and their relationship to the structural position of Black people. Du Bois argued that racial prejudice had significant deleterious consequences for Black people including (1) restricting Black people to menial/low-paying jobs; (2) displacing Black people due to competition from whites; (3) resentment of Black advancement and initiative; and (4) exposure to discourteous and insulting treatment in “social intercourse”, among others [2]. The contributions of Du Bois and other scholars have inspired contemporary scholars to investigate the health effects of Black Americans’ continual negotiation with white Americans about their assessments of self and place in society [3] (p. 13). 

For example, Farmer and Ferraro developed the diminishing returns hypothesis, in which they posit that Black–White health inequities are greatest at the highest levels of socioeconomic status [4]. While few studies have tested this hypothesis, there is a growing literature that has attempted to explain why Black–White health inequities are not completely explained by socioeconomic status (SES). Similar to Du Bois, contemporary scholars have investigated everyday measurements, navigation and negotiations as costs of upward social mobility [5,6,7,8]. Not only do these costs exact a toll on the quality of interpersonal interactions that Black Americans engage in over the life course, researchers have demonstrated that these costs likely diminish the social, economic and health returns on the human capital investments of Black Americans [7,8,9,10]. For example, Bell and colleagues found a greater obesity burden among African Americans with greater incomes (≥$100,000) and those who had completed college [11]. In accordance with the diminishing returns hypothesis, they argue that it is possible that African Americans experience fewer health-related benefits of greater levels of income and education compared with whites. 

The goal of this paper is to provide an assessment of the toll that African American men and women pay as they navigate predominately white spaces. The terms Black or African Americans are used interchangeably in this study. The paper provides a qualitative account of the potential health costs of upward social mobility and examines whether these costs diminish the health returns associated with additional socioeconomic resources. First, we describe racial heterogeneity in socioeconomic resources between Black and white Americans. Then we highlight the importance of chronic stress in the development of poor health and its contribution to Black–White health inequities. We then describe two unique stressors that upwardly mobile Black Americans face─ they must frequently navigate predominately white spaces and often play a critical supportive role in their social networks of origin. Our qualitative analysis describes the process and toll of navigating predominately white spaces among upwardly mobile Black Americans, which could help explain diminishing health returns on socioeconomic resources. Our analysis also investigates whether Black people face additional stressors in the form of social support and social capital demands from their networks of origin.

### 1.1. Racial Differences in Socioeconomic Resources

Wealth, such as net worth, net financial assets and home equity, is a fundamental resource that affects people across generations and across the life course [12,13]. Wealth protects against macroeconomic changes such as recessions, aids in weathering temporary financial crises such as unemployment and assists people with critical life transitions such as starting a family or purchasing a home [12,13]. The average Black household possesses approximately six to eight cents of wealth compared to every dollar of wealth held by the average white American household [13]. Although there have been gains in narrowing the income and education gap between Black and white people, the magnitude of the Black–White wealth gap has remained stubbornly entrenched [13,14]. 

This immense Black–White wealth gap is based upon historical legacies of racist policies and practices such as redlining, not individual choices such as frivolous spending or single parenthood [13]. Black Americans were systematically excluded from the practices that propelled the accumulation of white wealth and which continue to affect contemporary outcomes [14,15]. Specifically, New Deal era policies, a set of progressive, transformative laws developed in the aftermath of the Great Depression, increased the overall prosperity of Americans, including the transition to a majority homeowner society. However, Black Americans were excluded from wealth generating programs such as low-interest, federally backed home loans [15,16,17]. Due to redlining, i.e., the systematic denial of mortgages in majority Black neighborhoods, majority Black neighborhoods were largely excluded from this critical wealth generating mechanism [15,16,17]. 

As a result, the financial position of Black Americans, even those who are upwardly mobile, is often tenuous [17,18]. Furthermore, the dearth of wealth among Black Americans means that upward social mobility is largely dependent on tremendous personal effort and extensive investments in human capital, particularly attainment of advanced education and credentialing. African Americans may experience the stress of maintaining a palatable public identity in predominately white spaces in addition to strained social support networks, which may diminish their returns on human capital investments [5,7,8,9]. 

### 1.2. Chronic Stress and Health

A detailed description of the physiological stress reaction system is beyond the scope of this paper. However, in the face of an acute stressor, stress hormones such as epinephrine are released into the bloodstream, and a cascade of physiological changes, ranging from increased lung capacity and pupil dilation, are set in motion [19,20,21]. This response, often described as “fight or flight”, is critical in helping escape threats to one’s life. Researchers have demonstrated that chronic activation of this system is deleterious to health [21,22,23,24]. Yet, our bodies’ stress response systems are not sensitive enough to differentiate between social stressors, such as vigilance, and immediate threats to one’s life [25,26].

Chronic stress is more than just skin-deep, as researchers have documented that stressful experiences are deleterious to multiple health outcomes [26,27]. Chronic activation of the body’s stress response system leads to dysregulation over time, which is associated with multiple physiological changes such as poorer memory and emotional regulation and the development of atherosclerosis—the buildup of plaque in the arteries—and the storage of visceral fat [19,25]. These physiological changes, which accumulate over the life course, contribute to the development of chronic diseases such as cardiovascular disease and diabetes, which in turn affect overall rates of mortality and morbidity [23,28,29]. 

Researchers have documented that low social status is associated with chronic stress activation and posit that chronic stress exposure contributes to racial inequities in health [30]. For example, Geronimus’ weathering hypothesis is a theoretical framework that posits accelerated aging and premature physiological deterioration of Black women is due to frequent exposure to racism and socioeconomic disadvantage [30]. This process is hypothesized to contribute to poorer reproductive health outcomes [31,32,33]. Several scholars have found support for the weathering hypothesis, as there is evidence of elevated allostatic load (a measure of chronic physiological stress) scores among Black adults, compared to whites, which likely contributes to Black–White health inequities. As it relates to frequent navigation of white spaces, Hicken and colleagues have found that when people’s stress response is repeatedly activated due to apprehension over facing racism and discrimination (vigilance), dysfunction in the body’s regular stress response system occurs [33,34,35]. Even if an act of discrimination is not perceived, Black people remain hyper-alert and prepared to confront any instance of intentional or unconscious racism. Rumination over racist events can promote prolonged cognitive processing which can develop into a chronic stressor that repeatedly activates the stress response system. These experiences, in addition to overt, interpersonal discrimination, can have a negative effect on the health of African Americans. Scholars have documented that experiences related to racism affect a broad range of health outcomes, including mental health [36,37,38,39], self-reported health [40] sleep [33,41], cardiovascular disease [42] and more. Furthermore, while upwardly mobile African Americans may have greater SES, relative to Black Americans in their networks of origin, there are still tremendous racial gaps in income and wealth which likely contribute to the poorer health observed among African Americans [8,43,44,45]. 

### 1.3. Navigating White Spaces

Social mobility has been described as “movement which places an adult person into a social world that significantly differs from the one into which she or he was socialized during childhood” [46]. Due to the high levels of racial residential segregation in most American communities, the process of upward social mobility often requires that Black Americans must cross deeply entrenched racial boundaries into spaces that are predominately white [3,46]. Often, upwardly mobile Black Americans must make their way in predominately white spaces, especially in school and workplace settings. Findings from previous studies indicate that Black Americans with greater levels of SES are more likely to report exposure to racial discrimination than Black people with lower levels of SES [7,47,48,49,50]. 

Further, Black Americans are often highly visible in predominately white spaces. While educational institutions and companies take pride in maintaining the veneer of diversity, people of color are often called upon to “do diversity work”. This includes being asked to lead various diversity initiatives that are not valued or rewarded within organizations. For example, Evans and Leo Moore described the “impossible burdens” that people of color bear in predominately white spaces [51]. Scholars have found that upwardly mobile Black Americans expend considerable energy to fit into predominately white settings. Lacy argued that Black people must construct “public identities” to signal their class position and mitigate potential discriminatory experiences in public settings including at school, work or in stores [52,53]. Lacy’s respondents described broader efforts to signal social class and establish commonalities with white people in public spaces, labeling these strategies as “script switching”. This included using certain diction, wearing certain clothing, and discussing certain topics when interacting with white people. Lacy’s respondents simultaneously sought to disassociate themselves with negative stereotypes about Black people while also finding common ground with white people. Lacy noted, however, that the invocation of a public identity is a deliberate, conscious act that entails psychological costs and rewards. 

Similar to Lacy’s respondents, Sacks found that middle-class Black women devised strategies to present themselves in a manner that would not only establish social class commonalities with healthcare providers but also indicate their level of “health cultural capital”, or specific health related knowledge [10]. These were intentional “performances” used to gain respect and to avoid being viewed through the negative prism of stereotypes about Black women. Findings from a study conducted by McCluney and colleagues indicates that whites evaluated code-switching behaviors favorable and deemed efforts to conform as more professional [54].

The sum of these efforts to “perform” is exhausting. Findings from multiple papers indicate that effort to maintain a public face exacts a toll on Black Americans and leads to burnout [33,35,54]. Respondents in qualitative investigations of upwardly mobile African Americans report feelings of anger, hurt, disappointment and even rage as they struggle to outpace hegemonic, negative stereotypes and fight for professional legitimacy within white settings [55]. The toll is not limited to poor mental health. Researchers have demonstrated the stress of navigating predominately white spaces influences their likelihood to engage in health promotive behaviors such as physical activity. For example, Ray found that middle-class black men were less likely to be physically active in neighborhoods that are perceived as predominately white [56]. 

### 1.4. Upward Social Mobility & Uplift Stress

High levels of racial residential segregation reduce the availability and quality of resources, including access to social capital [57,58,59,60]. Segregation also means that Black Americans often must leave their social networks in order to pursue upward social mobility. For example, in her ethnographic study, Lareau reported that middle-class African Americans had less frequent contact with members of their extended family and lived further away from their extended families [60]. Therefore, many Black Americans must navigate the pathway to upward social mobility largely on their own. While their networks of origin provide encouragement and care, they often lack the experience and social connections that could aid in the negotiation of critical elevators to social mobility [3]. For example, Higginbotham found that Black working-class parents often did not possess the material (i.e., financial) and informational support that could help propel their children’s desires and plans for higher education [3]. Their broader social networks lacked the personal experiences of navigating the transition to college and beyond. In addition, Higginbotham found that respondents reported a great amount of pressure to determine key pathways to success. They also had to decipher unspoken social norms and expectations embedded in social settings that were very different from their own cultural and economic backgrounds. 

In addition to the lack of material and informational support that could be helpful in the pursuit of upward social mobility, previous studies indicate that upwardly mobile Black Americans feel responsible for the well-being of their entire social network. Scholars have documented that upwardly mobile Black people feel a great deal of internal pressure to give back to their networks of origin [61,62]. This includes the provision of informational support and social capital as well as the remitting of funds back to network members who have fewer socio-economic resources. Cole and Omari argued that while economic privilege may provide security and well-being to African Americans of greater SES, this accomplishment is often accompanied by some guilt and grief with respect to those left behind in poverty [5,6,61]. Heflin and Patillo (2006) found that middle-class African Americans are highly likely to have low-income siblings and that middle-class African Americans incorporate the socioeconomic status of their extended families into their own conceptions of class standing [62]. Higginbotham and Weber (1992) found that working class and middle-class African American women felt that they owed their family members for the help they received [62]. 

These findings not only demonstrate that Black Americans of higher socioeconomic status maintain strong links to their social support network, but also that their unique roles may be an additional stressor. Closeness to members of social networks could simultaneously be a coping resource and stressor as upwardly mobile African Americans may feel pressure to provide material support along with social capital to their networks of origin.

### 1.5. Present Study

Upward social mobility appears to come at a health cost to Black Americans [5,46,47,48,49,50]. There is a growing literature that has documented diminishing health returns on human capital investments [4,63,64]. The goal of this paper is to provide a qualitative account of potential health costs of upward social mobility and examine whether these costs diminish the health returns associated with additional socioeconomic resources. We felt that upwardly mobile would be the best way to describe this sample due to their own family backgrounds in addition to the lack of wealth in general in the African American community. This lack of wealth means that middle-class African Americans have often engaged in a process of upward social mobility, through hard work and investments in human capital, in order to achieve middle-class status.

## 2. Methods 

### 2.1. Sociohistorical Context of Race and Place in the St. Louis Area

St. Louis was selected as the data collection site to investigate the experiences of Black Americans especially in light of the uprising that occurred in Ferguson (a suburb of St. Louis) in August 2014. There are historical and contemporary racial tensions in the St. Louis area, which resulted in a tinderbox ignited by the outrage over the tragic shooting death of 18-year-old Michael Brown on 9 August 2014 by a white police officer in Ferguson. The events in Ferguson sent shockwaves throughout the St. Louis metropolitan area and across the United States, helping to galvanize the Black Lives Matter movement. The Ferguson uprising of 2014 represents an important turning point in the U.S.’s long history of racial animus, police violence, and class struggle. 

### 2.2. Data Collection and Recruitment

Data collection took place between November 2016 and August 2017. Focus groups and surveys were the primary data collection methods for the study. Focus groups were selected as the primary data collection method because they are useful for gleaning contrasting views of subgroups as well as determining which views are common across groups [65]. Respondents were included in the study if they identified as African American or Black, were 24 years or older and had completed college. Although other social class indicators such as income and wealth are important, education provided the most accessible screening tool for recruitment in our formative research. For example, in our pre-testing of our study protocol, respondents did not recall how much wealth they possessed. Similarly, it was more difficult to create income categories in recruitment materials. Additionally, education may be a better indicator of social class than income as education could reflect tastes, such as leisure time activities, styles of dress and dining preferences. We recruited through electronic distribution of flyers to various organizations such as Greek-lettered sororities and fraternities, as well as social media posts and word of mouth. Potential participants called the study coordinator, who determined eligibility and scheduled participants for a focus group over the telephone. The focus groups were held in the evening between 6-8pm at a well-known community agency in St. Louis.

Before the focus groups began, research assistants welcomed participants, reviewed the informed consent process and answered questions about the study. Participants also completed a 39-item demographic survey. This survey collected items related to household income, level of education, marital status, and home ownership status. On average, the survey took approximately 15 minutes to complete. 

This writer, a Black middle-class man (lead author of this paper) and a Black middle-class woman (second author) facilitated the focus groups with the support of graduate research assistants who took detailed notes. Race and gender matching focus group facilitators has been documented as a best practice in qualitative research as matching has been shown to improve rapport and promote more authentic responses [66]. Focus group questions included but were not limited to the following: “Describe some of the stressors that you experience?”; “How would you describe your responsibility to people in your network, community, extended family?”; and “Who do you turn to for support?” Focus groups lasted an average of 90 min. Each session was audio recorded and professionally transcribed. Participants provided verbal and written consent for participation prior to the start of the focus groups. Participants received $50 as an incentive for their participation in the study. The Institutional Review Board at Washington University in St. Louis approved this study.

### 2.3. Sample 

We conducted a total of six focus groups, three with men and three with women for a total of 32 respondents. Each group had 3–6 participants. The total number of male respondents was 12. The mean age for men was 39 (range = 26–57). Among men, all respondents reported that they had completed college and seven of the men reported that they had graduate or professional degrees. Five men reported household incomes of $80,000 or more; two reported household incomes of $65,000–$80,000 and 3 reported household incomes between $40,000–$55,000. One participant was in the $25,000–$40,000 range and one respondent refused to answer the income question. Regarding home ownership, four men reported that they currently owned homes while all other participants rented. Nine participants reported that they currently worked full-time while two reported part-time work and one reported that he was retired. Six participants reported that they were currently married or living with a partner. Three were single (never married) and three participants reported that they were divorced. 

Looking at the men’s focus group composition, two focus groups were comprised of three men while one group was comprised of six. The composition of the first group was six respondents with an average age of 41. In the first focus group of six, four participants reported that they were married and currently employed full-time. Three respondents in the first group reported household incomes of $80,000 or more while two reported incomes of $65,000–$80,000 and one man reported $40,000–$55,000. Three respondents reported that they had bachelor’s degrees while the other three reported that they had completed graduate or professional school. In the second group, the average age was 35 and all respondents reported that they had completed graduate or professional school and were employed full-time. Two of the men reported household incomes in the $40,000–$55,000 range while one reported $80,000 or greater. The average age of men in the third group was 41. Two men reported that they were currently employed full-time while one reported that he was a graduate student. Two of the men reported that they had completed graduate or professional school while one was currently enrolled in a graduate program. 

We conducted three focus groups with twenty college-educated, Black women. The groups were comprised of six to eight participants each (two focus groups were comprised of seven women while one group was comprised of six). The overall mean age was 33 years of age (range = 24–59). All participants had completed college and ten of the women reported that they had graduate or professional degrees. Examining subjective social class, five participants considered themselves upper middle-class, eight respondents identified themselves as middle class, six reported that they were lower-middle class, and one respondent identified herself as lower class. Regarding income, three women reported household incomes less than $40,000. Seven women reported household incomes of $40,000–$55,000; six women reported household incomes of $55,000–$80,000 and four women reported household incomes of $80,000. About home ownership, eight participants reported that they currently owned homes while all other participants rented or lived with family. Eighteen participants reported that they were employed full-time. One woman reported that she was a student and another reported part-time employment. Eleven women reported that they were currently single (never married) and five women were married/partnered. Three were divorced and one participant reported that she was separated. The average household size was two. 

Looking at the group level characteristics of the women’s groups, two focus groups were comprised of seven women while one group was comprised of six. The composition of the first group was eight respondents with an average age of 36. In the first focus group of seven, there were three women with college degrees and four women with graduate/professional degrees. Six participants reported that they were currently employed while one indicated that she was a full-time student. There were three single women, two married/ partnered women and two divorced women in the first group. Three respondents in the first group reported household incomes of $80,000 or more while three reported incomes in the $40,000–$55,000 range. One woman reported a household income in the $55,000–$80,000 range. In the second group, there were seven participants and the average age was 29. Three respondents reported that they held college degrees while four had completed graduate or professional school. Six women reported that they were employed full-time while one reported that she was employed part-time. Six women indicated that they were single while one reported that she was currently separated. There were two women who reported household incomes less than $40,000, four women who reported incomes between $40,000–$55,000 and one woman who reported her income between $55,000–$80,000. In the third group of six participants, the average age of the women was 39. All six indicated that they were currently employed full time. Four women indicated that they had completed college while two held graduate/professional degrees. Four women reported household incomes between $55,000–$80,000. One woman reported earning $80,000 or more and another reported less than $40,000. Two women reported that they were single, one woman was divorced and three were married. 

### 2.4. Analysis

The study followed the general theoretical assumptions of grounded theory as outlined by Charmaz. Charmaz’s (2006) constructivist approach advocates the use of sensitizing concepts, which provide conceptual grounding while remaining open to emergent themes [67]. As such, the study relied on an inductive approach to analyzing the data. Four analysts, trained graduate students, used sensitizing concepts gleaned from previous pertinent empirical research to develop the focus group questions and to guide subsequent data analysis. First, the analysts reviewed the transcripts against the audio-recordings to ensure accuracy. Second, the analysts met several times as a group to discuss their preliminary sets of codes and memos (margin notes that summarize emergent analytic themes). Analysts then identified text segments, coded those segments, and sorted them to identify higher-order themes [67]. To increase agreement between the coders, they attempted to develop consensus about the meaning of text. Agreement was based on whether the general passage was understood the same way, not whether the exact same words were highlighted. The team met again to decide on which themes represented higher-order concepts. Once a final list of codes was developed, analysts independently coded the transcripts and met once again to reach consensus on how the codes were applied. Analysts coded the transcripts independently using NVivo software (NVivo 12 2018). NVivo was used to manage and sort the data, which facilitated the comparison of themes across and within focus groups. Based on the analysis of men and women in this study, we arrived at three findings: (1) hypervisibility due to being the “diversity initiative” and subsequent vigilance and perceived pressure to represent all black people in white spaces; (2) uplift stress associated with upward mobility; and (3) health costs of upward mobility.

## 3. Findings

### 3.1. Hypervisibility—“I Am the Diversity Intuitive”

“‘We want someone with your perspective’…so beyond my daily work for my job email, there’s an influx of we need your—we want your perspective. And do you say no? Well, I don’t know, I’m the only one, you know. Uh, so there’s a level of stress of being responsive, um, not only to people you directly report to, but people within the institution you work for who see—who are trying, you know, in their good efforts to be more open and more inclusive, don’t necessarily recognize in that effort, you’re also placing more stress on me.”Ahmir, 37-year-old healthcare administrator

A consistent observation across all focus groups was that both men and women noted that in workplace settings, they are frequently viewed as “the diversity initiative”. As Ahmir noted above, this theme indicated that participants were often the only or one of a few black people in their respective places of employment. Like Ahmir, other respondents reported feeling pressure to represent other Black people who were not at the table as well as feelings of discomfort and isolation as they navigated predominately white workplaces. Respondents in this study reported that they were frequently called upon by their white colleagues to provide their perspective on race-related issues. 

For example, Kim, a 32-year-old engineer, noted that while her status as a Black woman in the science, technology, engineering, and math (STEM) field made her an unusual asset to her employer, she still was not equitably compensated: 

“…at my company, I’m the triple standard. I’m a Black woman who’s technical, who’s an engineer. That hasn’t necessarily given me more money—I know that, because we know statistically, they want me to feel like the unicorn.” 

In Kim’s case, although she is proud of her professional accomplishments, she describes being taxed in her workplace as she is expected to represent other Black people. Moreover, Kim described the social isolation she faced as one of only a few Black people, particularly Black women, at work. She stated: “I don’t have any allyships at work”.

Respondents often felt pressure, both internally and externally, to represent other Black people particularly at work. In the specific context of this study, just a few years after the shooting death of Michael Brown and subsequent uprisings throughout the St. Louis region, respondents were asked by their white colleagues to explain Black people’s perspective. Some were subject to awkward attempts to check in on their well-being during the Ferguson protests. For example, Audrey, a 29-year-old employed in federal law enforcement, noted:

“And every Black question that comes up, I’m the go-to person. Why do Black people do this thing? Oh, your hair is always different. Why do Black people do this thing? When Ferguson happened, why do Black people do this and what’s the whole issue? I’m not your go-to expert on Black. I don’t even understand half the stuff I do most day. I can’t speak for the entire Black community.”

Audrey’s experiences appear to be exhausting and uncomfortable. As just one of a few Black employees, she conveyed feelings of isolation. Audrey and others noted that their white colleagues frequently made inappropriate jokes or racist comments. However, respondents lamented that they had to be careful when determining which comments to respond to in order to disrupt negative stereotypes or to strive for a more inclusive environment. Respondents expressed a great deal of reticence in confronting white colleagues about their behavior because they were keenly aware of racist troupes, such as being an angry Black woman.

“I work in law enforcement which being a Black person in law enforcement is just not good right now ...it’s 99 percent white. Just the comments that I hear because I’m kind of more insular with the team. A lot of them are cops or former cops. One guy is telling this story about this is what I used to do when I had to chase this guy down and knocked his head up against here. And I’m sitting there like, I’m the only Black person at this table, and I’m like what part of this is okay that you feel like you can not only just say it in general but say it while I’m sitting here. That alone just made me step back and like, wait, this is too much. But then just walking around the offices, it’s just random comments that you just hear in conversations that are not necessarily at you, but it’s like really? I’m here. And I can’t say anything because I love my job, and I don’t want to lose my job over being perceived as being too sensitive because somebody questioning (sic) about being Black. But then at the same time it’s like can’t sit back and be quiet while you disrespect my blackness. So, it’s horrible.”

As Audrey notes, she is reluctant to confront her coworkers when they make racially offensive comments because she does not want to be perceived as overly sensitive, which may ultimately risk losing her job. 

In addition to being called upon to do diversity work in various organizations and agencies, respondents described other taxing experiences in which they were asked to defend their presence at work or school. For example, Marcus, a 30-year-old researcher, recounted a story about having his qualifications questioned in a social setting by a fellow graduate student. 

“We were having this party. In the middle of the party, she starts talking about this graduate assistantship that I got, and basically, she doesn’t understand how I got it over her. I’m really thinking like, you know, ‘Is this – is this about to go down right now? Are we about to have this, like, affirmative action talk at a party?’ Um, and yeah, it – it turned into us having this hour-long affirmative action talk.” 

Marcus noted how uncomfortable he felt, not only because he had to defend his qualifications, but also because he was the only Black person in the room. Despite this being an informal, social setting, the interaction became a confrontational space, which would affect Marcus’ future interactions with white colleagues. 

Participants also expressed the vulnerability they felt from being so isolated in their workplace or educational settings. 

“I’m the only Black person in that room, doing the work that I’m doing. And so, this comment slips out, and they feel comfortable saying it, but they also don’t recognize that someone who has a direct, adverse feeling towards that is right behind them. So, I literally say, ‘What?’ like, ‘What’s that?’ out loud, because that’s my defense, but I also recognize I’m the only Black—I’m the youngest, I’m a Black woman sitting in this role. I’m not a manager. I’m here as an assistant to the director, so it’s not like I have weight, anyway.”Kim, 32, engineer

Study participants noted that they had attempted to report instances of bias to their supervisors or through the appropriate reporting channels at work. Nonetheless, reporting specific instances of discrimination or bias was not often treated as a serious manner or nothing came of their reports. Black women participants reported experiences related to gender discrimination and bias as well. 

“…I’m on the elevator with one Black guy and he’s just ‘Mm, mm, mm.’ And I have to curse him out professionally and be like, ‘What is wrong with you? Is something wrong? Are you okay? Like why are you doing that?’” 

She continued:
“Walking down the hallway, ‘I like your dress.’ You don’t need to like my dress. Like I shouldn’t have to worry about what I’m wearing that day that if anybody else was wearing, you wouldn’t dare say that to them. So, when you have a certain look or you’re of a certain age or a certain body or whatever and people just feel like they can get away with certain things. And then you can go to anyone because I had the same thing, ‘Well, they’re just being nice’ or ‘That’s just Tom.’ I’m like I don’t care…that’s inappropriate.”Regina, 28, social worker

These comments illustrate the intersection of race and gender for upwardly mobile Black women. Black women shared experiences of navigating predominately white spaces in addition to confronting sexual harassment from black male colleagues. Furthermore, Black women voiced concerns about reporting harassment perpetrated by Black men because of a sense of camaraderie or linked fate with other black people. As such, they were reluctant to “tell on” Black men particularly at work. 

Vigilance, or feeling the need to remain watchful, was a consistent theme across all focus groups. As noted above in respondents’ experiences, negative interactions with white peers and colleagues were not perceived as one-time instances that were quickly forgotten. Rather, respondents perseverated on these experiences and carried them with them over time. Although respondents felt the need to observe their surroundings at all times, these experiences also contributed to respondents’ feeling that they were also being surveilled. Furthermore, respondents had to contort themselves, measuring their reactions to derogatory comments by white colleagues. Study respondents in both the men’s and women’s groups commented on the importance of physical presentation, often curtailing their own self-expression in order to be taken seriously in predominately white spaces and to avoid stereotypes. Respondents noted that they were received very differently by whites if they were not dressed in a manner that they deemed acceptable by whites, such as wearing a ball cap or braids in their hair. 

Respondents described a general feeling of unease at having to manage their own persona and signal their class status in an effort to make white people comfortable around them. 

“…when I’m in a professional environment, whether it be work or a conference or something, I also feel an expectation to represent black people in the best light and to be opposite of all the negative stereotypes.”Kevin, 51, lawyer

Because they knew they were under the spotlight in white spaces, they took care to curate their public identities. In addition, these comments reflect respondents’ desire to resist negative stereotypes. For most respondents, this resistance took the form of actively avoiding stereotypes by changing the way they spoke or dressed. For others, it was avoiding certain foods at company gatherings. Kevin, for example, noted that if he were at dinner with white colleagues, he would avoid watermelon and chicken. 

Other respondents, however, rejected these sentiments and voiced their attempts to resist the pressure to conform to perceptions of a “good Black person”. Ellis, a 26-year-old graduate student, stated the following:

“I don’t know—good, wholesome Black person in a white crowd, then you’re basically masking yourself for white people, right? This idea of coding is no longer a thing anymore for us, right? To code yourself and to be this ulterior black person, to be this, this wholesome Black person isn’t a thing for, at least, me anymore.”

He continued:
“I was taught to be this way, like my parents taught me to be—you know, ‘Don’t have braids.’ You know, all these things like that seem to be very much portrayed as stereotypical Black person, you know, the thug or in a gang... All that for me is gone. I don’t do that anymore because I know who I am. And so, for me, I’m not gonna be sitting here, trying to be all wholesome, like, ‘I’m a good Black man,’ if I’m not really like that…” 

Ellis’ comments reflect his awareness that Black people are often expected to manage their self-presentation and self-expression when in predominately white settings. He has been socialized to defy stereotypes in the company of whites. Even in the focus group, other participants encouraged Ellis to consider managing his self-expression and behaviors around white people. Nonetheless, Ellis resisted the pressures to conform into, what he called, “a wholesome Black person”.

These data indicate that Black Americans must weigh their desires to advance in professional and educational settings against conforming to social norms that dictate Black behavior in public settings. Even for Blacks who do conform to what they perceive as acceptable norms, data from previous studies indicate that pathways to upward mobility, such as getting hired, promoted, equitably compensated or securing clientele, remain blocked by racism. Further, being the “only one” necessarily leads to more job scrutiny than if there were other black people, especially black women, present in the workplace.

### 3.2. Uplift Stress 

“I am the provider in my family. So, I take care of my mother and my younger sister and, for a while, my father and my sister and her son. So, like my stress financially isn’t because I can’t provide for myself. It’s because I have to provide for other people.”Aja, 27-year-old woman employed in the service industry

A recurrent theme that emerged from these data is uplift stress, the pressure and responsibility to provide social and financial support to networks of origin. For example, Aja describes the pressure she feels to provide financial support to her family, including her parents and siblings. Participants reported uplift stress across all focus groups. This stress was exacerbated by respondents’ own financial status, which, despite having higher levels of education, was often unstable. Indicative of the lack of wealth in this population, participants reported accruing large amounts of debt, including student loans, credit card and business-related debt, to achieve their mobility goals. Yet participants also reported difficulty maintaining employment that was commensurate with their qualifications. Therefore, participants found it challenging to meet all their financial responsibilities, notwithstanding additional pressure to provide support to their networks of origin. Data from this study indicated that respondents did not just work toward their own personal and professional goals. Rather, they were concerned about the well-being of their entire network of origin and many of their decisions were driven by the desire to provide for their networks, particularly their parents. Steve, a 27-year-old engineer, remarked: 

“…my stress come from helping my parents get to where they will be okay, you know? I mean, um, being able to get it good enough so I can support them, if I need to.”

Participants felt that they could not refuse to help their families and other network members. For example, Marcus describes how he is perceived by members of his social network: 

“…there are a lot of people in my circle or network think that, you know, ‘Oh, Marcus never has any kind of, like, financial issues. And so, you know—you know, it’s gonna be—so basically, you know, I can just ask for whatever and—and he’ll able to—to provide that.’ In most cases, I mean I have provided, but it is, you know, sometimes at a detriment to myself, um, in a way. So, I’m, you know, giving money during times that I probably shouldn’t be giving money and that sort of thing.”

Although Marcus acknowledges that requests for money can be detrimental to his own financial stability at times, he is in a better financial position than many people he knows or with whom he grew up. 

Relatedly, participants such as Marcus felt that they could not make requests for financial assistance or other forms of social support. As Marcus indicated above, respondents felt that there was a misalignment between network perceptions and expectations and their own socioeconomic realities. Study participants indicated that they did not want to be an additional burden to network members who were struggling or who were, comparatively, in more stressful situations. This made them reticent to share their own stress or seek support. Another source of misalignment between participants and their networks was the lack of familiarity with experiences respondents were going through. For example, respondents reported difficulty in describing the challenges in their day-to-day lives, such as navigating white spaces or strategizing to achieve their mobility goals, to network members who had never been in similar situations. 

Women also described particularly challenging demands for support. Participants described the cumulative stress they experienced over time as they were asked to accommodate requests for support from their networks. One respondent quipped that she had recently lost her job and it provided a reprieve from the support requests because network members assumed that she did not have any money. Numerous participants reported that they intentionally make it uncomfortable for members of their network to ask for support to discourage additional requests in the future. Traci described her efforts to resist such requests: 

“If you were to call me maybe ten years ago, I struggled with it. I was—if I picked up the phone, if you needed this, okay, I would put it in your bank account tomorrow. If you need that, I’ll do this and I’ll do that. Probably in the last—I’m going to say last four years when I had my own child, now it’s, ‘No. I don’t have it. I’ll see. You all need to budget your money better.’ Cause at this point, you know, I’m looking out for my own child. I’m building her future.”Traci, a 38-year-old employed in management

### 3.3. Health Costs of Upward Social Mobility

“I’m depressed all over again, but I don’t have a network. That’s one of the things I think that is an added stressor for me since I moved here is I moved into place where I have no family, no friends, and I’m constantly thinking that if something happens back home or something happens here, nobody can get to me sooner than eight hours driving. [T]hen making friends at 30 years old when you’re not college and you’re in an all-white workplace.”Amanda, 29-year-old federal government employee

Amanda’s experience presents a comprehensive assessment of the health costs related to upward social mobility among Black Americans. She had to move to a new city for the purposes of upward social mobility, which left her isolated. Yet, she is also concerned about the well-being of her network of origin, now located over 500 miles away. Additionally, she works in a predominately white workplace, which leaves her feeling even more isolated.

Black men reported similar challenges but with the added pressure of conforming to norms of masculinity. For example, James and Ron noted their mental health struggles but indicate that stigma against seeking treatment is a barrier they must overcome. 

“For me it’s anxiety. Like, I think often—well, not think, but often I’m—I’m anxious about having to be a tough guy.”James, 39-year-old state government employee

“Recently, um, I’ve, uh, been concerned about my mental health, and I think for black males, the stigma of mental health and seeking help in that is, um, is proportionately, um, you know—they don’t—they don’t go seek help. So, for me and my stress, I—I decided to see somebody, meet a therapist and talk with somebody, um, to kind of talk through these things.”Ron, 36-year-old youth development director

The observations above are indicative of the toll Black Americans pay to attain middle-class status. Overall, participants described the costs of mobility, which may ultimately diminish the health returns related to increased socioeconomic resources. A consistent theme that emerged across the groups was feelings of exhaustion when navigating predominately white spaces. Many respondents attributed the exhaustion to being required to do diversity work at their places of employment while others reported being worn down by the pressure to construct and maintain an acceptable public persona. Additionally, respondents described the deleterious effects of conflicts over whether to confront white colleagues when they made racist comments, knowing that directly rebuking these comments would likely be uncomfortable and could also affect their professional reputations or place their jobs on the line. The pressure to defy stereotypes on a near daily basis, including efforts to conform, along with diminished returns on these efforts, is demoralizing and exhausting for upwardly mobile Black Americans. These experiences likely affect the health and well-being of Black Americans and could diminish health returns associated with greater socioeconomic resources. 

In addition to feelings of exhaustion and vigilance from being hyper visible while navigating predominately white spaces, respondents in this study lacked adequate social support to cope with stressful experiences. Social support is a critical coping mechanism as the enlistment of social support is salubrious. However, the findings from this study indicate that upwardly mobile black Americans may feel more isolated and have a great deal of demands on their resources from their social networks. Participants in this sample described the pressures from their social networks as stressful. While notions of social support are subjective, these data indicate that there is at minimum discomfort from attempting to satisfy the demands for support from social networks. Not only did study participants describe experiences of being isolated in workplace and educational settings, they did not have adequate social support to help cope with external stressors. 

## 4. Discussion 

Although socioeconomic status helps to narrow racial disparities in health, these inequities do not completely disappear, even when levels of education and income are similar between African Americans and whites [8,68,69,70,71,72]. The barriers to upward social mobility for African Americans are formidable. Due to lack of wealth, the financial position of upwardly mobile African Americans is often tenuous and most African Americans who achieve upward social mobility do so through extensive investments in human capital, particularly attainment of advanced education and credentialing, and hard work [5,6,9,13]. Furthermore, data from previous studies indicate that upwardly mobile African Americans feel responsible for the well-being of their entire social network and remit funds back to network members who are not socially mobile, which could truncate the advancement of upwardly mobile African Americans and serve as an additional stressor. African Americans must also contend with negative stereotypes and high levels of racial residential segregation throughout the U.S., which often requires that African Americans must constantly navigate predominately white spaces to achieve upward social mobility. 

The data from this study indicate that there are health costs associated with upward social mobility among Black Americans. Beyond experiencing interpersonal discrimination or overt forms of racism, respondents in this study reported stress due to being hyper visible in white spaces, especially in workplace settings. Participants reported that they felt vigilant, often anticipating unfair or biased treatment. They also felt pressure to represent other black people and to actively resist negative stereotypes. Although respondents noted that members of their social networks were proud of their success, they also felt pressure to provide financial and other resources to friends and family. Participants felt responsible for the well-being of their networks and did not want to refuse requests from network members even if providing support presented a hardship for participants. Pressure to provide support to participants’ networks was not only an additional stressor but seemed to diminish the social support available to them to cope with external stressors. 

This paper advances the literature forward in several meaningful and innovative ways. First, scholars have underscored the importance examining the joint effects of race/ethnicity, gender, social class and other aspects of social identity simultaneously [5,73,74,75,76]. This study provides an intersectional account of stressors related to diminished returns. Second, existing quantitative measures do not adequately capture the experiences of contemporary racism nor the discomfort of navigating white spaces when ambiguous incidents of interpersonal discrimination are not perceived or reported. Qualitative studies are needed, not only to document the phenomena of upwardly mobile Black Americans, but also to provide fodder for the eventual creation of new measurements that can better capture experiences of contemporary racism. 

### Limitations

While there are several assets in this study, we acknowledge some limitations. First, these data were collected in the post-Ferguson St. Louis area, a region with a specific racial and socioeconomic context. Although the findings are likely applicable to Black people in the U.S., and perhaps globally, we acknowledge the unique milieu in which the study was conducted. Second, we relied on educational attainment, specifically completion of college, as the primary marker of middle-class status in this study. While this sample was, on average, relatively highly educated, scholars may argue that other aspects of social class such as occupational prestige or income categories may be more appropriate inclusion criteria for the study. 

## 5. Conclusions

Upwardly mobile Black Americans are understudied, and the process of upward social mobility may not lend itself to the improved health outcomes that upwardly mobile whites experience. Because Black Americans have less capital on the front end, their investments in education, for example, may be diminished, characterized by lower compensation and slower advancement than whites [4,11,70,71]. The unique position of upwardly mobile Blacks may exacerbate stress-related conditions and processes due to decreased social support and increased exposure to racism that occurs in a variety of settings. This study helps to provide a more nuanced understanding of the negotiations that upwardly mobile Black Americans face and highlights social factors that could help explain diminished health returns.

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
