# Peer review of "The Price of the Ticket: Health Costs of Upward Mobility among African Americans"

_ijerph, 2020, doi:10.3390/ijerph17041179_

Round 1

Reviewer 1 Report

This manuscript described a study of the stressors related to upward mobility among college-educated black women and men living in St. Louis, MO. This is an important topic and the authors found that respondents described stress related to helping family and the feeling of being hypervisible and interactions with whites in the workplace. This qualitative study is a needed addition to the literature on black health. However, there are some needed improvements for the manuscript, particularly for publication in this particular journal.

First, the authors discuss Du Bois' double consciousness which is a necessary discussion in the black health literature, and follow with a description of racial inequities in wealth and its source, and how upwardly mobile blacks have to navigate white spaces. However, a lot more discussion of how these factors can affect black health. The authors mention diminished returns, but I was expecting some discussion of The Diminished Returns Hypothesis (see Farmer & Ferraro, 2005; Nuru-Jeter et al, 2019; Colen et al, 2018) or black health among those who are middle class or high SES (see Bell et al, 2018; Wilson et al, 2017; Jackson & Cummings, 2011; Thomas, 2015). Even if the authors do not want to use the Diminishing Returns Hypothesis to frame their argument, there needs to be a much stronger connection between the experience of black upward mobility and black health outcomes. Additionally, the authors seem to only refer to effects of stress which could be considered a mental health outcome. Would it be possible to refer to the mental health costs of upward mobility among blacks? If the authors would like to refer to health costs in general, a link between uplift stress and other health outcomes should be described. The authors refer to blacks, black people, African Americans and black Americans. If there is some rationale for this, it should be stated. If there is not, then consistency in use of terms should be achieved. The manuscript title refers to the health costs of upward mobility, but its not clear that the participants were upwardly mobile. Additionally, the focus group questions do not ask directly about upward mobility. Also, in the Introduction, the experience of "middle class" and upward mobility blacks seems to be described interchangeably. Some delineation here seems necessary. In the discussion, the authors state that "this study indicate[s] that there are health costs associated with upward social mobility among black Americans". Again, the only health outcomes indicated in the data were stress and depression. "Health costs" seems to broad for the results of this study.  Also, the authors conclude that the study provides understanding of upwardly mobile blacks, but there should be a discussion of the implications of the results for the literature and public health, as well as future work. Lastly, there are some misspellings and clerical errors. However, most importantly, there are several missing citations of previous literature that need to be included. Below are some examples (but most of the discussion section requires citations of statements of fact or previous studies that are referred to): High levels of racial residential segregation reduce the availability and quality of resources, including access to social capital. Upward social mobility appears to come at a health cost to black Americans. There is a growing literature that has documented diminishing health returns on human capital investments. This paper advancing the literature forward in several meaningful and innovative ways. Scholars have also underscored the importance examining the joint effects of race/ethnicity, gender, social class and other aspects of social identity simultaneously. Although socioeconomic status helps to narrow racial disparities in health, these inequities do not completely disappear, even when levels of education and income are similar between African Americans and whites.

Author Response

We appreciate the helpful comments and suggested edits by the anonymous reviewers. We believe that we have substantially strengthened the manuscript by addressing the issues raised in the review. We have responded to comments in detail below.

Reviewer 1

First, the authors discuss Du Bois' double consciousness which is a necessary discussion in the black health literature, and follow with a description of racial inequities in wealth and its source, and how upwardly mobile blacks have to navigate white spaces. However, a lot more discussion of how these factors can affect black health.

We have added a paragraph outlining the pathways between social experiences, such as navigating white spaces, and health. We framed this using the stress and coping framework and provided a greater discussion about the effects of chronic stress on health over the life course.

The authors mention diminished returns, but I was expecting some discussion of The Diminished Returns Hypothesis (see Farmer & Ferraro, 2005; Nuru-Jeter et al, 2019; Colen et al, 2018) or black health among those who are middle class or high SES (see Bell et al, 2018; Wilson et al, 2017; Jackson & Cummings, 2011; Thomas, 2015). Even if the authors do not want to use the Diminishing Returns Hypothesis to frame their argument, there needs to be a much stronger connection between the experience of black upward mobility and black health outcomes.

We have reverenced the diminishing returns hypothesis into the introduction and incorporated most of the citations listed into our introduction and discussion. We thank the reviewer for highlighting this work.  In addition to the references mentioned here, we also cited relevant work by Pearson, Ray and others.

Additionally, the authors seem to only refer to effects of stress which could be considered a mental health outcome. Would it be possible to refer to the mental health costs of upward mobility among blacks? If the authors would like to refer to health costs in general, a link between uplift stress and other health outcomes should be described.

We have added greater clarity throughout the paper underscoring the role of chronic stress exposure in the development of deleterious health for both mental and physical outcomes. We included more examples of how stress affects physical health outcomes such as cardiovascular disease as well as health behaviors such as likelihood to engage in physical activity.

The authors refer to blacks, black people, African Americans and black Americans. If there is some rationale for this, it should be stated. If there is not, then consistency in use of terms should be achieved.

There terms black and African American are used interchangeably in this paper. There is a sentence to say this now found in a footnote.

The manuscript title refers to the health costs of upward mobility, but its not clear that the participants were upwardly mobile. Additionally, the focus group questions do not ask directly about upward mobility. Also, in the Introduction, the experience of "middle class" and upward mobility blacks seems to be described interchangeably. Some delineation here seems necessary.

The focus group guide and survey included information related to social mobility. We did not include all the study questions in this paper and have noted this in the methods section on page 6 line 235. Our survey included information about parental SES, including parental education and occupational status. Furthermore, many of the respondents were not born and raised into middle-class families, despite their own middle-class status as adults.

We inserted the following footnote: We felt that upwardly mobile would be the best way to describe this sample due to their own family backgrounds in addition to the lack of wealth in general in the African American community. This lack of wealth means that middle-class African Americans have often engaged in a process of upward social mobility, through hard work and investments in human capital, in order to achieve middle-class status.

Lastly, the term middle-class is debated in various scholarly circles. However, an in-depth discussion of various sociological theories of social class is beyond the scope of this paper.

In the discussion, the authors state that "this study indicate[s] that there are health costs associated with upward social mobility among black Americans". Again, the only health outcomes indicated in the data were stress and depression. "Health costs" seems to broad for the results of this study. 

While we understand the author’s comment, we disagree that there should be a dichotomization of health outcomes into mental or physical health. We feel it would be too narrow to only focus on mental health in the discussion of stress.  As noted above, we included more discussion of the stress process and the development of poor health outcomes in the updated draft.

Also, the authors conclude that the study provides understanding of upwardly mobile blacks, but there should be a discussion of the implications of the results for the literature and public health, as well as future work.

 Lastly, there are some misspellings and clerical errors.

We have closely reviewed the updated draft for clarity and to remedy any errors.

However, most importantly, there are several missing citations of previous literature that need to be included. Below are some examples (but most of the discussion section requires citations of statements of fact or previous studies that are referred to): High levels of racial residential segregation reduce the availability and quality of resources, including access to social capital. Upward social mobility appears to come at a health cost to black Americans. There is a growing literature that has documented diminishing health returns on human capital investments. Scholars have also underscored the importance examining the joint effects of race/ethnicity, gender, social class and other aspects of social identity simultaneously. Although socioeconomic status helps to narrow racial disparities in health, these inequities do not completely disappear, even when levels of education and income are similar between African Americans and whites. This study provides a qualitative analysis of data from 6 focus groups conducted among African American men and women related to the topic of upward mobility. The following points will help strengthen the manuscript.

We have added citations to support each of the statements above.

Reviewer 2 Report

This study provides a qualitative analysis of data from 6 focus groups conducted among African American men and women related to the topic of upward mobility. The following points will help strengthen the manuscript.

1) Lines 26-163: Could authors ensure that the introduction is succinct (currently it is almost 4 pages long)? Please, also, clearly identify the goal of the study. Is it “to provide an assessment of the toll that African American men and women pay as they navigate predominantly white spaces” (as indicated in lines 51-52)? Or is it more specific and that the goal is to explore how these “costs of upward social mobility” (line 154) “could diminish health returns” (line 155)? Please indicate what has been done in the literature for this specific goal so that what this study adds to the literature is stated concisely and is transparent to the readership.
2) Lines 165-173: Could authors describe the characteristics of the study setting from the perspective of the primary goal of this study?
3) Lines 174-203: Could authors describe the processes that were followed immediately after each focus group?
4) Lines 174-203: When were the focus groups conducted?
5) Lines 184-186: Could authors provide additional information about the recruitment process for the focus groups? Please especially discuss how this recruitment process ensured that a group of African American men and women with diverse views on the topic were reached for the focus groups.
6) Lines 190-192: This study seems to use a survey to collect sensitive, private information from focus group participants; survey is not an independent data collection instrument. Please emphasize here and in other parts of the manuscript (such as in line 175) that a survey instrument was used to help collect background information on focus group participants.
7) Lines 190-192: Please explain how the confidentiality of the private information collected in the survey was secured.
8) Lines 190-192: Could authors provide a copy of the 39-item demographic survey instrument as an appendix? Please also describe if this instrument was pre-tested?
9) Lines 193-194: How were the “middle-class men” who moderated the focus groups recruited? Were the moderators the same or different for each of the 6 focus groups? Why? How many graduate students attended each focus group?
10) Lines 196-199: Could authors provide a copy of the focus group questions guide? How many sections did the guide have; how many questions were included in each of the sections? Did authors piloted the guide?
11) Lines 205-230: Please provide a table with the sociodemographic information that is discussed here and others that has been collected via 39-item survey instrument (ensuring that small cells are suppressed).
12) Lines 205-206 & lines 216-218: Could authors discuss how they decided that a sample size of 32 (12 men and 20 women) was appropriate to reach their study goal?
13) Lines 205-206 & lines 216-218: Please be very specific and provide the number of participants for each of the 6 focus groups. Also, please reconcile “three focus groups comprised of six to eight participants in each group (two focus groups were comprised of seven women while one group was comprised of six)…”.
14) Lines 232-251: Could actors discuss if there was member checking?
15) Lines 232-237: While “remaining open to emerging themes” in the data is an inductive approach, could authors discuss if their use of knowledge from the previous empirical work on the topic as a guiding tool for “subsequent data analysis” for the study is also an “inductive approach”?
16) Lines 235-246: Please clarify who those “four analysts” are that are referred to here. Also reconcile with 3 analysts indicated in line 559.
17) Line 246: Please specify how many analysts “coded the transcripts independently” clarifying who they are.
18) Line 260: Please resolve the issue of confidentiality of focus group participant here and for all other quotes in the manuscript. Also, please reconcile with the reference to “anonymous research participants” in line 564.
19) Lines 253-401: Could authors clearly discuss what is meant by “hypervisibility”? Is it being isolated because they are one of the few or the only African American in their workplace? Or is it the stress of having to “represent other black people”? Or is it the experiences they have when they are faced with “inappropriate jokes and racist comments”? Or is it that their “qualifications” were “being questioned in a social setting”? Or is it how their reports of “bias to their supervisors” are treated? Or is it the “feelings of vigilance”? Or is it their efforts “to curate their public identities”?
20) Lines 518-521: Could authors provide references of these “previous studies”? Also, please indicate how the current study enhanced findings these previous studies?
21) Lines 562-563: Please be clear if this research was externally funded or not.
22) Lines 566-572: Please clearly state if there were any conflicts of interest or not.
23) Please ensure that acronyms (such as SEP in line 87) are spelled out the first time they are mentioned and that the citations (such as reference #s 8, 11 & 19) are complete.

Author Response

We appreciate the helpful comments and suggested edits by the anonymous reviewers. We believe that we have substantially strengthened the manuscript by addressing the issues raised in the review. We have responded to comments in detail below.

Reviewer 2:

Lines 26-163: Could authors ensure that the introduction is succinct (currently it is almost 4 pages long)?

While we would like to narrow the introduction according to the reviewer’s suggestion, these are complex issues that need to be properly fleshed out. Furthermore, as indicated by reviewer 1’s comments, there is the need to provide an even longer introduction.

Please, also, clearly identify the goal of the study. Is it “to provide an assessment of the toll that African American men and women pay as they navigate predominantly white spaces” (as indicated in lines 51-52)? Or is it more specific and that the goal is to explore how these “costs of upward social mobility” (line 154) “could diminish health returns” (line 155)?

We have simplified the study goal: Upward social mobility appears to come at a health cost to black Americans. There is a growing literature that has documented diminishing health returns on human capital investments.. The goal of this paper is to provide a qualitative account of potential health costs of upward social mobility and examine whether these costs diminish health returns related to improve socioeconomic resources.

Lines 165-173: Could authors describe the characteristics of the study setting from the perspective of the primary goal of this study?
3) Lines 174-203: Could authors describe the processes that were followed immediately after each focus group?

This is explained in the methods section as it stands. Participants were thanked for their time and we completed a debrief with participants.

Lines 174-203: When were the focus groups conducted?

We are not certain whether the reviewer is referring to the time of day or range of months that we collected data. We inserted the following sentences in the methods just to be sure.

Data collection took place between November 2016 and August 2017.

The focus groups were held in the evening between 6-8pm at a well-known community agency in St. Louis.

Lines 184-186: Could authors provide additional information about the recruitment process for the focus groups? Please especially discuss how this recruitment process ensured that a group of African American men and women with diverse views on the topic were reached for the focus groups.

This is covered extensively in the method section as it stands. And we are not a way to pre-select participants based on their views into a sample in general.

Lines 190-192: This study seems to use a survey to collect sensitive, private information from focus group participants; survey is not an independent data collection instrument. Please emphasize here and in other parts of the manuscript (such as in line 175) that a survey instrument was used to help collect background information on focus group participants. 

This is in the methods section:

Before the focus groups began, research assistants welcomed participants, reviewed the informed consent process, and answered questions about the study. Participants also completed a 39-item demographic survey. This survey collected items related to household income, level of education, marital status, and home ownership status, etc. On average, the survey took approximately 15 minutes to complete.

7) Lines 190-192: Please explain how the confidentiality of the private information collected in the survey was secured.

As mentioned, the study was approved by our institutional review board. In accordance with those policies, we de-identified all information in the survey and used pseudonyms for the focus group participants. We also stored all paper copies of materials under double lock and key. We do not think this is necessary to included in the body of the paper is not commonly included in most papers.

8) Lines 190-192: Could authors provide a copy of the 39-item demographic survey instrument as an appendix? Please also describe if this instrument was pre-tested?

No, we do not wish to include a copy of the survey in the appendix. The survey was pre-tested with graduate students prior to collecting data from participants and was comprised of validated measures.

9) Lines 193-194: How were the “middle-class men” who moderated the focus groups recruited? Were the moderators the same or different for each of the 6 focus groups? Why? How many graduate students attended each focus group?

The PI’s of the study conducted the focus group. There was one man PI and one woman PI. We stratified the groups by gender (3 groups each). There was one graduate student note taker in each group.

We specified this in the methods:

One black middle-class man and women (the lead authors for this paper), facilitated the focus groups with the support of graduate research assistants who took detailed notes.  

10) Lines 196-199: Could authors provide a copy of the focus group questions guide? How many sections did the guide have; how many questions were included in each of the sections? Did authors piloted the guide?

No, we will not provide a copy of the guide. As mentioned in the methods, the guide was piloted with graduate students prior to data collection with participants.

11) Lines 205-230: Please provide a table with the sociodemographic information that is discussed here and others that has been collected via 39-item survey instrument (ensuring that small cells are suppressed).

We have included a table to reflect the sociodemographics of the sample.

12) Lines 205-206 & lines 216-218: Could authors discuss how they decided that a sample size of 32 (12 men and 20 women) was appropriate to reach their study goal?

13) Lines 205-206 & lines 216-218: Please be very specific and provide the number of participants for each of the 6 focus groups. Also, please reconcile “three focus groups comprised of six to eight participants in each group (two focus groups were comprised of seven women while one group was comprised of six)…”.

14) Lines 232-251: Could actors discuss if there was member checking?

Although member checking is often used in qualitative research to enhance trustworthiness and credibility, we did not use it for this study due to logistical constraints, e.g., difficulty reaching participants again. Moreover, although it is an acceptable way to enhance rigor, some scholars have pointed out that the method presents ethical challenges, e.g., that participants may be shown transcripts without sufficient context thereby causing them harm (Brit et al., 2016).

15) Lines 232-237: While “remaining open to emerging themes” in the data is an inductive approach, could authors discuss if their use of knowledge from the previous empirical work on the topic as a guiding tool for “subsequent data analysis” for the study is also an “inductive approach”?

This is nonsensical.

16) Lines 235-246: Please clarify who those “four analysts” are that are referred to here.

There were four trained graduate students involved in the coding process. We included an explanation of this in the methods on page 8 line 299.

18) Line 260: Please resolve the issue of confidentiality of focus group participant here and for all other quotes in the manuscript. Also, please reconcile with the reference to “anonymous research participants” in line 564.

We mentioned that the given names in this manuscript are pseudonyms, a common practice in reporting qualitative findings. This is an acknowledgement of the participants. This is very straightforward.

19) Lines 253-401: Could authors clearly discuss what is meant by “hypervisibility”? Is it being isolated because they are one of the few or the only African American in their workplace? Or is it the stress of having to “represent other black people”? Or is it the experiences they have when they are faced with “inappropriate jokes and racist comments”? Or is it that their “qualifications” were “being questioned in a social setting”? Or is it how their reports of “bias to their supervisors” are treated? Or is it the “feelings of vigilance”? Or is it their efforts “to curate their public identities”?

This is explained in the manuscript.

Reviewer 3 Report

This research is thought provoking and timely.

Strengths: The author's use of first hand research, through focus groups to assess the health taxation on upward mobile African Americans provides much needed qualitative data in the area. Tis research is very timely and relevant, as African Americans upward mobility is increasing and growing in media representation. These personal accounts help give life to the authors larger argument, and provides much needed real-life accounts to the discourse on the physiological impact of academic and career advancement on African Americans.   

Areas for Improvement: The author relies heavy on pre-existing literature to set the tone for their larger argument; however, this literature review could benefit from empirical based examples that demonstrate the argument. For example, in the section on Upward Social Mobility and Uplift Stress, the author cites important literature on the topic, but does not provide any empirical evidence. Without the quantitative data, the argument appears antidotal. The author must also address literature that contradicts their claim. Addressing this literature will strengthen the larger argument. Finally, using quantified data in the results could be helpful to substantiate their claims. For instance, under Hypervisibility, the author states “A consisted observation across forces groups.” It may be helpful to give numerical counts when generalizing in this way.

Author Response

We appreciate the helpful comments and suggested edits by the anonymous reviewers. We believe that we have substantially strengthened the manuscript by addressing the issues raised in the review. We have responded to comments in detail below.

Reviewer 3:

Areas for Improvement: The author relies heavy on pre-existing literature to set the tone for their larger argument; however, this literature review could benefit from empirical based examples that demonstrate the argument. For example, in the section on Upward Social Mobility and Uplift Stress, the author cites important literature on the topic, but does not provide any empirical evidence.

We disagree. The papers covered are empirical. We did not rely solely on theoretical or conceptual papers to make these arguments.

Without the quantitative data, the argument appears antidotal. The author must also address literature that contradicts their claim. Addressing this literature will strengthen the larger argument. Finally, using quantified data in the results could be helpful to substantiate their claims. For instance, under Hypervisibility, the author states “A consisted observation across forces groups.” It may be helpful to give numerical counts when generalizing in this way.

We disagree and this comment reveals an apparent bias against qualitative work in general. These findings are consistent with other qualitative studies and help to explain findings drawn from quantitative studies. We are not counting responses. The quotes are meant to illustrate the larger themes that emerged.

Round 2

Reviewer 1 Report

The authors have addressed most of the comments given in the first review of the manuscript and have improved the manuscript greatly. The only additional concern is that the authors have suggested that they included a footnote as a response to the previous review and I cannot find it in the revised manuscript. 

Please see below:

We inserted the following footnote: We felt that upwardly mobile would be the best way to describe this sample due to their own family backgrounds in addition to the lack of wealth in general in the African American community. This lack of wealth means that middle-class African Americans have often engaged in a process of upward social mobility, through hard work and investments in human capital, in order to achieve middle-class status.

Author Response

We are thankful to the reviewer for pointing out this missing footnote. We have inserted the footnote in the updated draft on page 6 in the present study section. 

Reviewer 2 Report

Thanks to authors; this is an improved manuscript but, as indicated below, some of the reviewer points have not been addressed.

The following points need to be addressed. Old numbering is kept for convenience:

1) Old Lines 26-163: Thanks for adding to this section. With all due respect, this reviewer still considers a long introduction as putting the emphasis in the wrong place; away from the details of the study conducted. Also, could authors ensure to turn on the track changes so that old parts of the manuscript will not be highlighted? Additionally, yes, as authors indicate (on new line 79: “citation?”), providing a citation for any assertion made is best. Assertions with citations that specifically pertain to the subject matter is better.
2) Old Lines 165-173: Based on reviewer’s reading of the manuscript, the primary goal seems to be upward social mobility among African Americans and the “hypervisibility” and stresses/health costs this presents as described by the focus group participants. It will be helpful to provide the study setting description from this perspective.
3) Old Lines 174-203: Thanks for internal response. What was the aim of this “participant debriefing” and how did it help with increasing the quality of data collected for this study?
5) Old Lines 184-186: The reviewer understands that the authors do not need to “pre-select” but it would be best if they can reassure the readership that their findings come from a diverse group of participants.
8) Old Lines 190-192: The readership should be able to follow what was asked to participants before the focus group discussion, especially given the 39-item survey instrument. Please discuss how the administration of this survey (which was just used to aid in collecting sensitive information given no other goal has been reported so far) influenced the focus group data. Were there any questions in the instrument that focus group participants found especially sensitive?
10) Old Lines 196-199: Thanks for adding “… but were not limited to …” (new line 257) but the point in the first review was made to ensure that enough information is provided to the readership to enhance transparency and understanding.
11) Old Lines 205-230: The reviewer could not find a reference to the table in the manuscript. Please use track changes. How similar were the sociodemographic findings from the focus group participants to those with similar sociodemographic characteristics in the study setting?
12) Old Lines 205-206 & lines 216-218: The 2 (out of 3) focus groups with only 3 men each as participants are extremely limited and raises concerns.
13) Old Lines 205-206 & lines 216-218: Please explain the information contained in new lines 305-307 about focus group sizes for women.
14) Old Lines 232-251: Please use track changes to indicate where internal points provided about member checking was included in the manuscript.
15) Old Lines 232-237: The internal response is an unfortunate one. The authors should have discussed if they made any efforts to update/revise their guide, processes etc after what they learned from each focus group given the inductive approach they are following.
16) Old Lines 235-246: Thanks for adding “…trained graduate students…” (new line 330) but there is still a need to reconcile “four analysts” with 3 analysts indicated in new line 670.
17) Old Line 246: Please specify how many analysts “coded the transcripts independently” clarifying who they are.
18) Old Line 260: Thanks for using pseudonyms. Also, please discuss how including age and occupation of participants are not identifying information.
19) Old Lines 253-401: Could authors clearly discuss what is meant by “hypervisibility”? Is it being isolated because they are one of the few or the only African American in their workplace? Or is it the stress of having to “represent other black people”? Or is it the experiences they have when they are faced with “inappropriate jokes and racist comments”? Or is it that their “qualifications” were “being questioned in a social setting”? Or is it how their reports of “bias to their supervisors” are treated? Or is it the “feelings of vigilance”? Or is it their efforts “to curate their public identities”?
21) Old Lines 562-563: Please be clear if this research was externally funded or not.
22) Old Lines 566-572: Please clearly state if there were any conflicts of interest or not.
23) As indicated in the first review, please ensure that acronyms (there is a new one in this version on the new line 49) are spelled out the first time they are mentioned.

Author Response

Old Lines 26-163: Thanks for adding to this section. With all due respect, this reviewer still considers a long introduction as putting the emphasis in the wrong place; away from the details of the study conducted. Also, could authors ensure to turn on the track changes so that old parts of the manuscript will not be highlighted? Additionally, yes, as authors indicate (on new line 79: “citation?”), providing a citation for any assertion made is best. Assertions with citations that specifically pertain to the subject matter is better.

We are satisfied with the length of the introduction. The manuscript fits the journal space requirements and we addressed the comments from reviewer 1 by elaborating on some content in the revision. We feel that Reviewer 2’s editorial preference for a shorter introduction is not an a substantive critique, especially since the reviewer is simply stating that they believe the introduction is too long without any guidelines about what should be cut or how long it should be. 

We appreciate the reviewer highlighting the type for line 79. We simply missed adding the appropriate citation here. This has been corrected in the updated version.

Old Lines 165-173: Based on reviewer’s reading of the manuscript, the primary goal seems to be upward social mobility among African Americans and the “hypervisibility” and stresses/health costs this presents as described by the focus group participants. It will be helpful to provide the study setting description from this perspective.

This is adequately covered in the present manuscript.

Old Lines 174-203: Thanks for internal response. What was the aim of this “participant debriefing” and how did it help with increasing the quality of data collected for this study?

The debriefing was simply to check in with participants to see how they were feeling after the focus group. We also shared next steps about the study with participants. We did not collect data from this process.

Old Lines 184-186: The reviewer understands that the authors do not need to “pre-select” but it would be best if they can reassure the readership that their findings come from a diverse group of participants.

We are satisfied with the inclusion criteria and sampling and believe this information is clearly presented in the current manuscript. In a qualitative study, the goal is not to obtain a nationally representative sample or to gather as wide a range of perspectives as probability driving sampling. We were careful not to overgeneralize our findings for this reason.

8) Old Lines 190-192: The readership should be able to follow what was asked to participants before the focus group discussion, especially given the 39-item survey instrument. Please discuss how the administration of this survey (which was just used to aid in collecting sensitive information given no other goal has been reported so far) influenced the focus group data. Were there any questions in the instrument that focus group participants found especially sensitive?

This information is already adequately included in the current manuscript. We collected information in the survey to better characterize the study sample and to more efficiently collect these data.

10) Old Lines 196-199: Thanks for adding “… but were not limited to …” (new line 257) but the point in the first review was made to ensure that enough information is provided to the readership to enhance transparency and understanding.

We believe there is adequate information provided for readers to understand the questions and content covered during the focus group studies. What we have provided is aligned with the methods of previous qualitative studies.

11) Old Lines 205-230: The reviewer could not find a reference to the table in the manuscript. Please use track changes. How similar were the sociodemographic findings from the focus group participants to those with similar sociodemographic characteristics in the study setting?

The goal in a qualitative study is not provide a comparison of sociodemographic characteristics among study respondents. We did, however make reference to the table in the updated manuscript.

The sample is too small to accurately compare different individual participants. Second, unlike a quantitative study, the unit of analysis in a focus group study is not the individual. It is the group. We provided a great deal of detail describing each of the groups.

This also relates to the next comment:

12) Old Lines 205-206 & lines 216-218: The 2 (out of 3) focus groups with only 3 men each as participants are extremely limited and raises concerns.

Again, the unit of analysis is the group, not the individual. What we have conducted and presented is commiserate with previous qualitative focus group studies.

13) Old Lines 205-206 & lines 216-218: Please explain the information contained in new lines 305-307 about focus group sizes for women.

We are not sure what the reviewer means here. This section simply provides the group characteristics for each focus group. This is clearly stated in the manuscript.

14) Old Lines 232-251: Please use track changes to indicate where internal points provided about member checking was included in the manuscript.

We addressed this comment in the previous revision: Although member checking is often used in qualitative research to enhance trustworthiness and credibility, we did not use it for this study due to logistical constraints, e.g., difficulty reaching participants again. Moreover, although it is an acceptable way to enhance rigor, some scholars have pointed out that the method presents ethical challenges, e.g., that participants may be shown transcripts without sufficient context thereby causing them harm (Brit et al., 2016).

15) Old Lines 232-237: The internal response is an unfortunate one. The authors should have discussed if they made any efforts to update/revise their guide, processes etc after what they learned from each focus group given the inductive approach they are following.

It is unfortunate the reviewer did not clearly articulate their question/ concern. We did not adjust or revise our focus group guide over time based upon responses. Although some qualitative studies using grounded theory may revise the interview instrument over the course of data collection, that is done more typically with an in-depth interview guide used for a series of individual interviews. Given that the research design used focus group data collected over a short time period, we felt that keeping the focus group constant was appropriate. Moreover, based on the research team’s discussion of how well the questions worked, we did not see any reason to revise the questions. Finally, although it is possible and acceptable to revise questions, it is by no means a requirement to revise questions during the course of data collection.

16) Old Lines 235-246: Thanks for adding “…trained graduate students…” (new line 330) but there is still a need to reconcile “four analysts” with 3 analysts indicated in new line 670.

This has been reconciled in the text.

17) Old Line 246: Please specify how many analysts “coded the transcripts independently” clarifying who they are.

Four analysts coded the transcripts.

18) Old Line 260: Thanks for using pseudonyms. Also, please discuss how including age and occupation of participants are not identifying information.

Providing ages and occupations to describe study respondents, but not including their names or other identifying information, e.g., not disclosing if they have a very unusual or specialized profession, is standard practice in social science research, including qualitative research. For example, referring to someone as a social worker, engineer, or healthcare administrator, as we have done in the manuscript, in no way discloses their actual identity.

19) Old Lines 253-401: Could authors clearly discuss what is meant by “hypervisibility”? Is it being isolated because they are one of the few or the only African American in their workplace? Or is it the stress of having to “represent other black people”? Or is it the experiences they have when they are faced with “inappropriate jokes and racist comments”? Or is it that their “qualifications” were “being questioned in a social setting”? Or is it how their reports of “bias to their supervisors” are treated? Or is it the “feelings of vigilance”? Or is it their efforts “to curate their public identities”?

21) Old Lines 562-563: Please be clear if this research was externally funded or not.

22) Old Lines 566-572: Please clearly state if there were any conflicts of interest or not.

This is included in the journal materials and not commonly included in the manuscript portion.

23) As indicated in the first review, please ensure that acronyms (there is a new one in this version on the new line 49) are spelled out the first time they are mentioned.

We have spelled out socioeconomic status for SES in the updated manuscript.